


# Deep Learning Based Post-Process Correction of the Aerosol Parameters in the High-Resolution Sentinel-3 Level-2 Synergy Product

Antti Lipponen[1], Jaakko Reinvall[2], Arttu Väisänen[2], Henri Taskinen[2], Timo Lähivaara[2], Larisa Sogacheva[1], Pekka Kolmonen[1], Kari Lehtinen[1,2], Antti Arola[1], and Ville Kolehmainen[2]

[1]Finnish Meteorological Institute, Atmospheric Research Centre of Eastern Finland, Kuopio, Finland
[2]University of Eastern Finland, Department of Applied Physics, Kuopio, Finland

**Correspondence:** Antti Lipponen (antti.lipponen@fmi.fi)

**Abstract.** Satellite-based aerosol retrievals provide global spatially distributed estimates of atmospheric aerosol parameters that are commonly needed in applications such as estimation of atmospherically corrected satellite data products, climate modeling and air quality monitoring. However, a common feature of the conventional satellite aerosol retrievals is that they have reasonably low spatial resolution and poor accuracy caused by uncertainty in

auxiliary model parameters, such as fixed aerosol model parameters, and the approximate forward radiative transfer models utilized to keep the computational complexity feasible. As a result, the improvement and re-processing of the operational satellite data retrieval algorithms would become a tedious and computationally excessive problem. To overcome these problems, we have developed a machine learning-based post-process correction approach to correct the existing operational satellite aerosol data products. Our approach combines the existing satellite retrieval data

and a post-processing step where a machine learning algorithm is utilized to predict the approximation error in the conventional retrieval. With approximation error we refer to the discrepancy between the true aerosol parameters and the ones retrieved using the satellite data. Our hypothesis is that the prediction of the approximation error with a finite training data set is a less complex and easier task than the direct fully learned machine learning based prediction in which the aerosol parameters are directly predicted given the satellite observations and measurement

geometry. With our approach, there is no need to re-run the existing retrieval algorithms and only a computationally feasible post-processing step is needed. Our approach is based on neural networks trained based on collocated satellite data and accurate ground based AERONET aerosol data. Based on our post-processing approach, we propose a post-process corrected high resolution Sentinel-3 Synergy aerosol product, which gives a spectral estimate of the aerosol optical depth at five different wavelengths with a high spatial resolution equivalent to the native resolution of the

Sentinel-3 level-1 data (300 meters at nadir). With aerosol data from Sentinel-3A and 3B satellites, we demonstrate that our approach produces high-resolution aerosol data with better accuracy than the operational Sentinel-3 level-2 Synergy aerosol product or a conventional fully learned machine learning approach.



## 1    Introduction

Climate change is one of the biggest challenges our society is facing today (IPCC, 2021). Despite the rapidly
progressing climate research, projections of the future climate still contain large uncertainties with anthropogenic
aerosol forcing being among the largest sources of these uncertainties (Pachauri et al., 2014). If more accurate global
information about the athmospheric aerosol parameters such as the aerosol optical depth (AOD) and Angstrom
exponent (AE), and consequently of their product aerosol index (AI), were available, it would enable more accurate
modelling of anthropogenic aerosol forcing and could lead to a significant reduction of the uncertainties in future
climate projections. Another major challenge for our societies is air quality. In 2017, 2–25% of all deaths worldwide
were attributable to ambient particulate matter pollution (GBD 2017 Risk Factor Collaborators et al., 2018). To
monitor more accurately air quality and pollution sources near real time spatially high resolution estimates of aerosols
are needed (van Donkelaar et al., 2015).

Ground based aerosol observations can be obtained from the Aerosol Robotic Network (AERONET) which utilizes
ground based direct sun photometers (Holben et al., 1998). AERONET stations produce accurate information on
aerosols because they directly observe the attenuation of solar radiation without interference from land surface
reflections. However, AERONET has the limitation that the network consist of a few hundreds of irregularly spaced
measurement stations, leading to a very limited and sparse spatial coverage of aerosol information. The only way to
get wide spatial coverage information on aerosols is to use satellite retrievals.

Aerosol satellite retrieval algorithms produce estimates of the aerosol optical properties such as AOD given the
satellite observation data such as the top-of-athmosphere reflectances or radiances and the information on the
observation geometry. Satellite retrieval algorithms have been developed for multiple satellite instruments and the
available satellite aerosol data records span already time series that are over 40 years long (Sogacheva et al., 2020).
Examples of satellite aerosol data products include the Moderate Imaging Spectroradiometer (MODIS) aerosol
products (Salomonson et al., 1989; Levy et al., 2013), and Sentinel-3 Synergy aerosol products.

A satellite aerosol retrieval requires solution of a non-linear inverse problem, where the task is to find aerosol
parameters that minimize a misfit (such as the least squares residual) between the satellite observation data and a
forward model, which models the causal relationship from the unknown aerosol parameters to the satellite observation
data. Atmospheric monitoring satellites cover the globe almost daily with spatial high resolution observation data,
resulting in huge amount of daily data to be processed by the retrieval algorithms. Due to the excessive amount of
data, the operational aerosol retrieval algorithms employ physically and computationally reduced approximations of
radiative transfer models as the forward models (e.g. lookup-tables) and relatively simple inverse problem approaches,
which often ignore some of the observation data to reach fast computation times (Dubovik et al., 2011). Further, the
retrieval algorithms typically produce spatially averaged aerosol products that have lower spatial resolution compared
to the native satellite level-1 observation data. Because of these approximations and reductions, the aerosol retrievals
have limited accuracy and sub optimal spatial resolution.



Machine learning based solutions have been recently proposed for satellite aerosol retrievals in many studies. Compared to conventional inverse problems approaches, machine learning based solutions lead to much faster computation time (once the model has been trained) and they also offer a flexible framework for utilization of learning data based prior information in the retrieval. Most of the machine learning approaches to aerosol retrieval employ a fully learned approach where the machine learning model is trained to emulate the retrieval directly, that is, to predict the values of the unknown aerosol parameters given the satellite observation data (top-of-atmosphere radiances or reflectances) and observation geometry as the inputs. In Randles et al. (2017) neural network based fully learned aerosol retrievals are assimilated into NASA's MERRA-2 re-analysis model. In Di Noia et al. (2017), a fully learned neural network model is used to retrieve the initial AOD for an iterative retrieval algorithm. In Lary et al. (2009), a fully learned approach with MODIS retrieved AOD and the surface type as additional inputs was used for the AOD retrieval from MODIS data. The results of Lary et al. (2009) were validated using the accurate (AERONET) data Holben et al. (1998). The authors were able to reduce the bias of the MODIS AOD data from 0.03 to 0.01 with neural networks, while with support vector machines even better improvement was reported - AOD bias was less than 0.001 and the correlation coefficient with AERONET was larger than 0.99. However, they performed validation using all the available AERONET network stations both for training and validation. The split between the training and validation datasets was carried out using random sets of the MODIS pixel values. With the random split of all pixels, the data samples from the same AERONET station were present both in training and evaluation datasets, leading potentially to overfitting as the model learns, for example, the surface properties at the locations of the AERONET stations and can thus predict the aerosol properties very accurately at these locations but may not generalize well to data from other regions. In Albayrak et al. (2013), a neural network based fully learned model was trained and evaluated for MODIS AOD retrieval. In their model, MODIS reflectances, measurement geometry information, MODIS AOD and its quality flag were used as the input to predict the AOD. They found their model to produce more accurate AOD retrievals than the operational MODIS Dark Target algorithm. In Lanzaco et al. (2017), a slightly different type of machine learning based approach was used to improve satellite AOD retrievals. The authors used MODIS AOD retrievals and local meteorology information as inputs to predict the AOD in South America. This approach that combines the conventional AOD retrievals and local meteorology information was found to improve the AOD accuracy over the operational MODIS AOD. A problem in fully learned approaches is that they rely only on the training data and do not employ physics-based models in the retrievals. This may cause problems for the model to generalize to cases in which the inputs are outside the input space spanned by the training dataset.

In Lipponen et al. (2020) we proposed a model enforced machine learning model for post-process correction of satellite aerosol retrievals. The key idea in the model enforced approach is to exploit also the model and information of the conventional retrieval algorithm and train a machine learning algorithm for correction of the approximation error in the result of the conventional satellite retrieval algorithm. Previously, the post-process correction approach has been found to produce more stable and accurate results than a fully learned approach in generation of surrogate simulation models (Lipponen et al., 2013, 2018) and in medical imaging, see for example Hamilton et al. (2019). The





advantages of the model enforced post-process correction approach are improved accuracy over the existing data products and fully learned machine learning approach, and the possibility to post-process correct existing (past) satellite data products with no need for full recalculation of the retrievals. In Lipponen et al. (2020), the model
enforced approach was combined with a Random Forest regression algorithm for post-process correction of MODIS AOD and AE products using collocated MODIS and AERONET aerosol data for training the correction model for the approximation error in AOD and AE in the MODIS DT over land product. The post-process correction was found to yield significantly improved accuracy over the MODIS AOD and AE retrievals, and the correction approach resulted in better accuracy retrievals than the fully learned machine learning approach.

In this paper, we propose a post-process corrected high resolution Sentinel-3 Synergy aerosol product. The product is based on the high resolution Sentinel-3 level-2 Synergy land product aerosol parameters with 300 meter spatial resolution and the model enforced machine learning approach, where a feed forward neural network is trained for post process correction of the approximation error in the Sentinel-3 level-2 Synergy aerosol product. The training of the neural network is based on collocated Sentinel-3 Synergy and AERONET data from five selected regions of
interest. Given the Sentinel-3 observation data and high resolution aerosol products as input, our model produces an estimate of the AOD at five wavelengths utilizing the native 300m resolution of the Sentinel-3 observation data.

The rest of this paper is organized as follows. In section we describe the approximation error model for post-process correction of the satellite aerosol retrieval. Section 3 explains the preprocessing of the Sentinel-3 and AERONET data for machine learning and the neural network model used for the regression task. Section 4 gives the results and
Section 5 the conclusions.

## 2    Post-process correction model of satellite aerosol retrievals

Let $\mathbf{y} \in \mathbb{R}^m$ denote an accurate satellite aerosol retrieval

$$\mathbf{y} = f(\mathbf{x}), \tag{1}$$

where vector $\mathbf{y}$ contains the output of the satellite retrieval algorithm, $f : \mathbb{R}^n \mapsto \mathbb{R}^m$ is an accurate retrieval algorithm
and $\mathbf{x} \in \mathbb{R}^n$ contains all the algorithm inputs including the observation geometry and level-1 satellite observation data such as the top-of-atmosphere reflectances. Typically, the retrieval is carried out one image pixel at a time and the aerosol retrieval $\mathbf{y}$ can consist, for example, AOD and AE for a single image pixel, or as in the present study, AOD in a single image pixel at five wavelengths.

In practice, due to uncertainties in the auxiliary parameters, such as land surface reflectance, of the underlying
forward model utilized in the retrieval, extensive computational dimension of the problem and processing time limitations, it is not possible to construct an accurate retrieval algorithm $f$ but an approximate retrieval algorithm

$$\tilde{\mathbf{y}} \approx \tilde{f}(\mathbf{x}) \tag{2}$$



has to be employed instead. The approximate retrieval $\tilde{f}$ is typically based on physically simplified and computation-
ally reduced approximate forward models that are used due to huge amount of data and the need for computational
efficiency. The utilization of the approximate retrieval algorithm leads to an *approximation error*

$$e(\mathbf{x}) = f(\mathbf{x}) - \tilde{f}(\mathbf{x}) \tag{3}$$

in the retrieval parameters.

The core idea in the model enforced post-process correction model is to improve the accuracy of the approximate
retrieval (2) by machine learning techniques Lipponen et al. (2020). By Equations (1)-(3), the accurate retrieval can
be written as

$$
\begin{aligned}
\mathbf{y} &= f(\mathbf{x}) \\
&= \tilde{f}(\mathbf{x}) + \left[ f(\mathbf{x}) - \tilde{f}(\mathbf{x}) \right] \\
&= \tilde{f}(\mathbf{x}) + e(\mathbf{x}).
\end{aligned} \tag{4}
$$

To obtain the corrected retrieval, Equation (4) is used to combine the conventional (physics based) retrieval algorithm
$\tilde{f}(\mathbf{x})$ and a machine learning based model $\hat{e}(\mathbf{x})$ to predict the realization of the approximation error $e(\mathbf{x})$ to obtain
a corrected retrieval

$$\mathbf{y} \approx \tilde{f}(\mathbf{x}) + \hat{e}(\mathbf{x}). \tag{5}$$

Note that this approach is different from a conventional fully learned machine learning model in which the aim is to
emulate the accurate retrieval algorithm $f(\mathbf{x})$ with a machine learning model

$$\mathbf{y} \approx \hat{f}(\mathbf{x}) \tag{6}$$

that is trained to predict the retrieval $\mathbf{y}$ directly from the satellite observation and geometry data $\mathbf{x}$, see Figure 1
for a flowchart of fully learned and model enforced regression models.

The reason why the model enforced approach (5) can be expected to perform better than the fully learned model
(6) is that the approximation error $e(\mathbf{x})$ is a simpler function for machine learning regression than the full physics-
based retrieval $f(\mathbf{x})$ thus resulting in more accurate results than with a fully learned approach Lipponen et al.
(2013, 2018). Also, while the fully learned approach utilizes an ensemble of satellite observation data as learning
data, the model enforced approach utilizes also the additional information in the approximate retrievals. Also, as
the training of the post process correction is based on existing satellite data and retrievals, the implementation can
be done in a straightforward manner, for example, using black-box machine learning code packages and used for
correction of past satellite retrievals without recomputing the approximate retrieval products $\tilde{f}(\mathbf{x})$. In addition, the
post process correction model is also flexible with respect the choice of the statistical regression model, and the
choice of the regression model can be tailored to different retrieval problems separately.





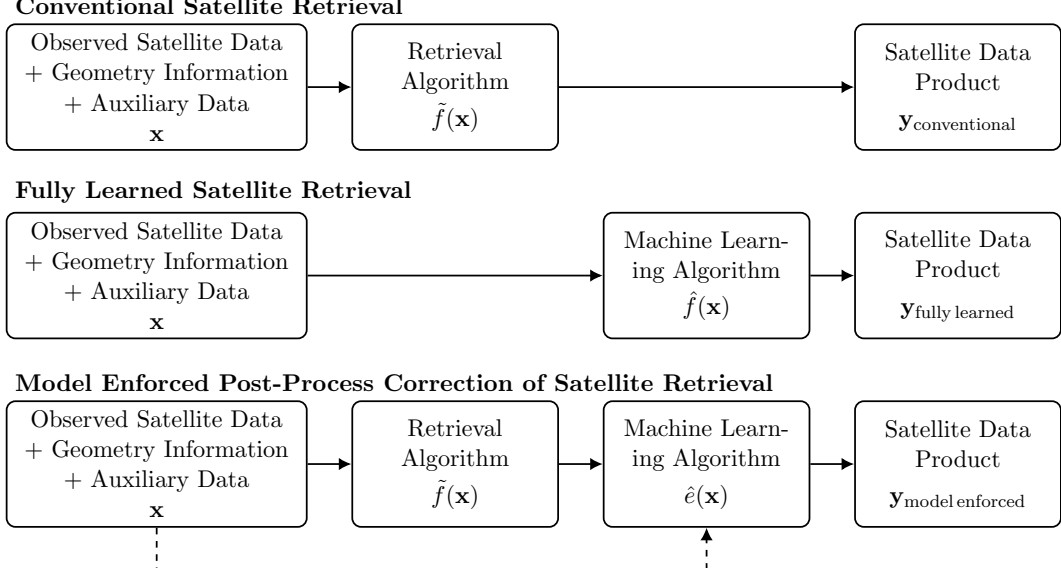

**Figure 1.** Top: Conventional satellite retrieval. Middle: Fully learned machine learning based satellite retrieval approach. Bottom: Model enforced post-process correction satellite retrieval approach.

## 3 Methods

This section describes the construction of the learning and test data for the machine learning retrieval of Sentinel-3 aerosol product with the post-process correction model (5) and the fully learned model (6). The selection of the neural network models and training of the networks is also described. For training and validation of the post-process correction, we use the high resolution Sentinel-3 level-2 Synergy and AERONET aerosol data.

### 3.1 Sentinel-3 satellite datasets

Sentinel-3 is a European ocean and land mission. Currently two satellites related to this mission (Sentinel-3A and 3B) are flying and collecting data. In this study, we use the Sentinel-3 Ocean and Land Color Instrument (OLCI) and Sea and Land Surface Temperature Radiometer (SLSTR) data. OLCI is a medium-resolution imaging spectroradiometer (spatial resolution about 300 m at nadir) with 21 spectral bands from 400 to 1020 nm. SLSTR is an imaging radiometer with dual-view capabilities. The pixel size of SLSTR is from 500 meters to 1 km and spectral coverage is from visible to thermal infrared in 9 standard bands (S1-S9). The swaths of these two instruments overlap allowing combined products that exploit data from both instruments. The high resolution Sentinel-3 level-2 Synergy land aerosol product is this type of combined product which we will post-process correct by the model (5).

We use both level-1b and level-2 data of the Sentinel-3 satellite mission data products from both Sentinel-3A and Sentinel-3B satellites. The level-1b data includes the information about the measurement geometry and the satellite





observed reflectances. The level-2 data includes the Synergy retrieval data and the corresponding quality information.

We use the SLSTR level-1b data from the product `SL_1_RBT`, OLCI level-1b data from the `OL_1_ERR` data product and Sentinel-3 level-2 data from the `SY_2_SYN` data product. We use year 2019 data in our study. For more information on the Sentinel-3 mission datasets, see https://sentinel.esa.int/web/sentinel/missions/sentinel-3/data-products. The Sentinel-3 data used in the models are listed in the appendix.

## 3.2 AERONET

AERONET is a global network of sun photometers Holben et al. (1998). AERONET has a Direct Sun data product that has both the AOD and AE data that we will use for training and testing of the machine learning models. AERONET is commonly used as an independent data source and all the data is publicly available at the AERONET website (http://aeronet.gsfc.nasa.gov/). An extensive description of the AERONET sites, procedures and data provided is available from this website. Ground-based sun photometers provide accurate measurements of AOD, because

they directly observe the attenuation of solar radiation without interference from land surface reflections. The AOD estimated uncertainty varies spectrally from ±0.01 to ±0.02 with the highest error in the ultraviolet wavelengths Eck et al. (1999). In this study, we use AERONET, Version 3, level-2, Direct Sun algorithm data. The AERONET variables used in our studies are listed in the appendix.

## 3.3 Regions of interest

The training and testing of the post process correction model is based on Sentinel-3 and AERONET data for year 2019 from five regions of interest shown in Figure 2. The regions of interest were selected so that different types of aerosol regions based on aerosol source and type, AOD values and different types of surface reflectances are included and also that the areas have good enough coverage of AERONET stations.

The data for the machine learning procedures consist of collocations of Sentinel-3 pixels with aerosol information

and AERONET data. We use similar collocation procedure as in Petrenko et al. (2012) but with reduced radius of 5 km. We also require that the aerosol data in the pixels we use is not flagged as filled, climatology data, too low values, high error, partly cloudy or ambiguous clouds. Furthermore, we require that the pixels we use do not contain any cosmetic level-1 data. Our selections lead to a total number of 5526 collocated Sentinel-3 - AERONET overpasses for the machine learning procedures.

The AERONET stations were divided to separate training, validation and testing sets for good generalization of the machine learning procedures. More specifically, the stations were randomly split into two sets for two-fold cross validation. To ensure as equal spatial distribution of AERONET stations as possible in both sets, we carried out the random split separately for each region of interest.





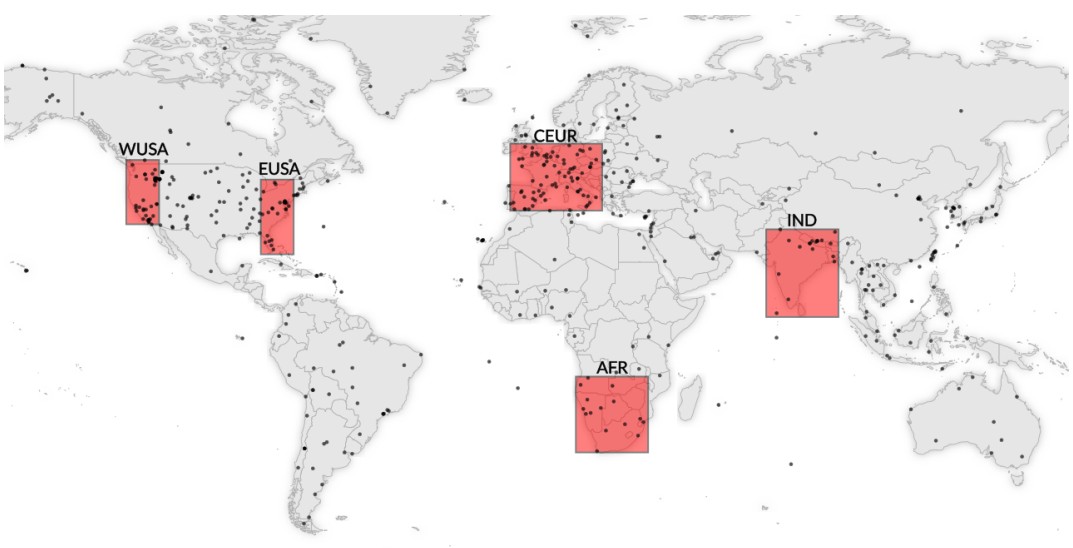

**Figure 2.** Regions of interest. Black dots indicate locations of AERONET stations.

### 3.4 Input and output data for the machine learning models

The aerosol retrieval $\mathbf{y} \in \mathbb{R}^5$ in both, the post-process correction approach (5) and the fully learned approach (6), consist of AODs for a single $300 \times 300 \text{m}^2$ (at nadir) image pixel at wavelengths 440nm, 500nm, 550nm, 675nm and 870 nm. These wavelengths are native wavelengths in the AERONET and Sentinel-3 level-2 Synergy aerosol products in the sense that the AERONET produces AOD at 440nm, 500nm, 675nm and 870nm and the Synergy product at 550nm.

In the fully learned model (6), the regression target $\mathbf{y} \in \mathbb{R}^5$ consist of the AERONET AODs at the selected five wavelengths. The AERONET AOD at the Synergy 550nm channel was estimated as the mean of AOD 550nm obtained from Angstrom law based on AERONET AOD at 500nm and AE 440-870nm. The input data for the fully learned model contains Sentinel-3 satellite geometry and observation variables for a single image pixel. All the input and output variables were standardized by subtracting the training data set mean and dividing by the standard deviation. To retain the spectral dependency of the AOD values at different wavelengths, all the AOD variables were standardized together using the mean and standard deviation of all AOD wavelengths. In case some of the inputs contains a missing value, it is filled with the average value of the training dataset. We also add a binary (0/1) inputs for each input variable to indicate if the data was filled. These selections and processing leads to an input vector $\mathbf{x} \in \mathbb{R}^{90}$. See the appendix for the Sentinel-3 data file variable names of the the inputs and outputs.

In the post-process correction approach, the regression target $\mathbf{e} \in \mathbb{R}^5$ consist of the approximation error between AERONET and Synergy spectral AOD. The Synergy aerosol product contains AOD and AE at 550 nm, which are transformed by the Angstrom law to obtain the Synergy AOD product at the wavelengths 440nm, 500nm, 675nm





and 870nm. The input data of the post-correction model contains the same geometry and level-1 data variables that are used in the fully learned model plus the Sentinel-3 level-2 Synergy aerosol data. Furthermore, the inputs and outputs are standardized and the missing values filled similarly as for the fully learned model. These selections lead to an input $\mathbf{x} \in \mathbb{R}^{156}$.

### 3.5 Deep learning based regression models

A fully connected feedforward neural network was selected as the model for the supervised learning tasks of estimating the regressors $\hat{f}(\mathbf{x})$ in (6) and $\hat{e}(\mathbf{x})$ in (5). In the neural network, the rectified linear unit (ReLu) was used as the activation function for all the hidden layers and no activation function was employed for the output layer. The weight coefficients of the neural net were estimated by minimization of the MSE loss functional with the ADAM optimizer. In the network training, batch size was 512, initial learning rate $5 \cdot 10^{-5}$ and the termination criteria for the learning was set to maximum 10000 epochs or until validation loss started to increase with patience tolerance set to 10 epochs. For further information on deep learning and neural networks, see e.g. (Goodfellow et al., 2016).

The architechture of the feedforward neural networks were optimized by utilizing the Asynchronous Successive Halving Algorithm (ASHA) (Li et al., 2020). In the ASHA optimization the maximum number of trial network architectures was set to 2500 and the algorithm was allowed to use up to 500 epochs in a single trial. The space of feasible states for the number of hidden layers in the ASHA optimization was set to $(2, 3, 4)$ and the number of nodes in the hidden layers was allowed to be up to the number of elements in the input vector $\mathbf{x}$. The optimization of the network architechtures by ASHA led to the network structures shown in Figure 3 for the fully learned approach $\hat{f}(\mathbf{x})$ and the post-process correction approach $\hat{e}(\mathbf{x})$. These network acrhitectures were utilized in the final training of the models.

### 3.6 Implementation

The neural network computations were implemented in Python utilizing Pytorch and the ASHA optimization utilizing the Ray-tune package. The codes for the fully learned model and post-process correction model will be made available. See the code and data availability for information on how to obtain the code to run the post-process correction and load a sample dataset.

## 4 Results

The accuracy of the post-process correction is tested using AERONET data as the ground truth for the aerosol retrievals and the results are compared to the high resolution Sentinel-3 level-2 Synergy aerosol product and to the fully learned retrieval model (6).

Figure 4 shows scatter plots of the AOD retrievals with the Sentinel-3 level-2 Synergy product (left column), fully learned machine learning (middle column) and post-process correction model (right column) against the AERONET





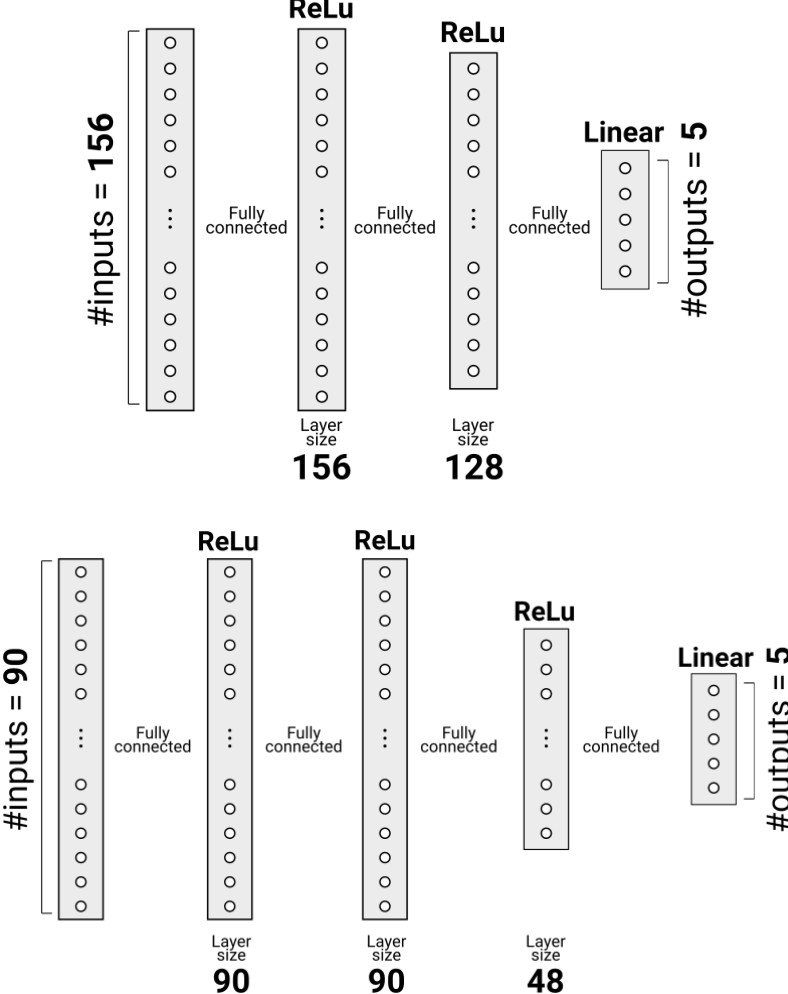

**Figure 3.** Schematic figure of neural network architectures used. Top: Correction network $\hat{e}(\mathbf{x})$. Bottom: Regression network $\hat{f}(\mathbf{x})$.

data at all the test data stations at the four visible to near infrared wavelengths 440nm, 500nm, 675nm and 870nm

measured by the AERONET. Each figure shows the coefficient of determination based correlation coefficient $R^2$, root mean squared error (RMSE), and median bias as the metrics to compare the retrievals. The figures show also the ratio of samples that are inside the Dark Target over land expected error (EE) envelope of $\pm(0.05+15\%)$. As can be seen, the machine learning approaches improve clearly the accuracy of the AODs compared to the high resolution Sentinel-3 level-2 Synergy product. Between the two machine learning approaches, the post-process correction model

has otherwise better $R^2$, RMSE and median bias error metrics than the fully learned model with the exceptions of the bias being the same as with the fully learned model at 500nm and 675nm. The ratio of samples inside the Dark





Target EE envelope are very similar with the post-process correction and fully learned models. A notable feature in the figures is that there are significantly less samples and relatively more "outliers" for large AOD values than for small AOD values. The accuracy of the machine learning estimates also improves for the higher wavelengths, which do contain fewer high AOD values. These findings can be attributed to the fact that the learning data contains relatively few samples for large AOD (the number of samples with AOD>0.5 is less than 5%). This indicates that more high AOD value learning data would be needed to improve the prediction of the high AOD values.

Figure 5 shows comparison of AOD at the native Sentinel-3 level-2 Synergy wavelength 550nm, AE and AI. Given the estimated AODs at the five wavelengths, the AE was estimated as a separate post processing step by utilizing the standard approach (e.g. in AERONET) where AE is estimated by a least squares fit to the linearization of the Angstrom law. In AERONET, the AE estimation is carried out using ordinary least squares type of method that rejects clear outliers from the data to improve the outlier tolerance of the AE estimation. The difference to AERONET AE obtained using ordinary least squares fitting with no outlier treatment, however, is small. The aerosol index (AI) is computed then as product of the AOD and AE. AI has been considered as a better proxy for cloud condensation nuclei (CCN) than AOD (Gryspeerdt et al., 2017), since AI is more sensitive than AOD to the accumulation mode aerosol concentration. Figure 5 shows that the machine learning approaches lead to clearly improved estimates of AOD 550nm, AE and AI compared to the Sentinel-3 level-2 Synergy product. The post process correction approach produces the best RMSE, $R^2$ and EE metrics for the AOD estimates. From the AE estimates, we observe that the high resolution Sentinel-3 level-2 Synergy AE product is uninformative as it produces the same constant value (approximately 1.1) for all of the test data points with a wide range of AERONET AEs. For the AE, the post-process correction approach has smaller bias and visibly better correlation (with nearly two times larger $R^2$ metric) but worse RMSE than the fully learned model. For the AI the post-process correction has better RMSE, bias and $R^2$ metrics compared to the fully learned model.

Figure 6 shows AERONET and Sentinel-3-based time series of AOD at 550 nm over three AERONET stations, Madrid, Paris, and Rome_Tor_Vergata for year 2019. In all stations, the overestimation of AOD by the Sentinel-3 level-2 Synergy product is evident. The Sentinel-3 level-2 Synergy AOD has also a clear seasonal cycle with higher AODs occuring on summer and lower AOD on winter. Both the fully learned model and post process corrected Sentinel-3 Synergy AOD are in very good agreement with the AERONET AOD. Furthermore, the regressor and post process correction model AOD capture very well the events of elevated AOD with duration of some days.

In Figure 7, monthly averages of AOD at 550 nm in western Europe for January, April, July and October 2019 are shown for the Sentinel-3 level-2 Synergy, fully learned model and post-process correction model based data. Again, the significantly higher AOD of Sentinel-3 level-2 Synergy compared to the other two models is evident. The figure also clearly shows that the amount of data varies quite significantly throughout the year mainly due to clouds and snow and more data is available for April and July than January and October. All datasets show some spatial variations of AOD over Europe and some cities and regions, such as Paris, France and Po Valley, Italy, clearly show up in AOD maps.





Figure 8 shows monthly averages of AOD at 550 nm for Madrid, Paris and Rome in July 2019. The filled circles in the images indicate the monthly averages of the AERONET stations present in the regions. The Sentinel-3 level-2 Synergy data product clearly produces a much higher AOD values then the fully learned and post-process correction models, and the overestimation with respect to AERONET is also evident. The Sentinel-3 level-2 Synergy AOD is also, due to spatial median filtering of the data, much smoother than the two other models. For the fully learned and post-process correction models, the AOD values are very close to the AERONET AODs at the AERONET sites and some high-resolution features are also clearly visible in the data. For all three cities, both the fully learned and post-process correction model show some neighbourhoods with elevated AOD. The correction model AOD shows even more details and less artefacts than the fully learned model AOD. For example in Rome, the road from the city center to the airport is clearly visible from the AOD data while the regression model does not show this road. The fully learned model also has some more box-shaped spatial anomalies than the other models.

## 5 Conclusions

We have developed a deep learning based post-process correction of the aerosol parameters in the high resolution Sentinel-3 level-2 Synergy land product. Sentinel-3 Synergy has also an aerosol data product specifically designed to retrieve the aerosol parameters. The aerosol data product, however, has spatial resolution of 4.5 km whereas the land product provides data with the Sentinel-3 instrument's full spatial imaging resolution of 300 meters. The drawback in the Synergy land product aerosol parameters is their relatively poor accuracy. The aim of the post-process correction is to significantly improve the accuracy of the Sentinel-3 level-2 Synergy land product aerosol parameters. The correction is carried out as a computationally light-weight post-processing step and therefore there is no need for re-running the actual Synergy retrieval algorithm to obtain the corrected aerosol data. As a reference for the machine learning based post-process correction of the Sentinel-3 level-2 Synergy data product we also trained a fully learned machine learning based regression model that carries out the full aerosol retrieval using Sentinel-3 level-1 data.

The results show that the fully learned and post-process correction machine learning approaches produces a clear improvement in the aerosol parameter accuracy over the official Synergy data product. The post-process correction approach leads generally to a more accurate aerosol parameters than the fully learned approach. The post-process correction approach combines information both from the physics-based conventional retrieval algorithm and machine learning correction whereas the fully learned model does not include any physics-based model information. The inclusion of the physics-based model information may make the post-process correction approach more tolerant against samples outside the range of the training data set when compared to the fully learned approach. The results show that the fully learned model results more often in high errors than the post-process correction.

The high spatial resolution, about 300 meters at nadir, and the high accuracy of the post-process corrected Sentinel-3 Synergy aerosol parameters over the official Sentinel-3 level-2 Synergy data product may possibly enable usage of



the data for new applications. For example, for air quality applications, the high resolution accurate aerosol data could be a step towards street level monitoring instead of the typical city or neighbourhood levels in conventional aerosol data products. Improved accuracy high spatial resolution aerosol parameter information may significantly also benefit atmospheric correction in many land surface satellite applications. The most impacted land surface applications are especially those that retrieve information from very low signal to noise ratio data such as the

retrieval of vegetation solar-induced fluorescence.

*Code availability.* Python code and trained models to run the post-process correction are available at https://github.com/ TUT-ISI/S3POPCORN

*Video supplement.* Video corresponding to Figure 7 can be found online at https://doi.org/10.5281/zenodo.5287243

## Appendix A: Sentinel-3 data used

This section describes the Sentinel-3 data used in the study. We use both level-1b and level-2 data of the Sentinel-3 satellite mission data products and we use data from both Sentinel-3A and Sentinel-3B satellites. For more information on the Sentinel-3 mission datasets please see https://sentinel.esa.int/web/sentinel/missions/sentinel-3.

### Level-1b

### SLSTR

We use SLSTR level-1b data from the `SL_1_RBT` data product. The variable names and the corresponding filenames in the data products are listed in Table A1.

### OLCI

We use OLCI level-1b data from the `OL_1_ERR` data product. The variable names and the corresponding filenames in the data products are listed in Table A2.

### Level-2

### Synergy

We use Sentinel-3 level-2 data from the `SY_2_SYN` data product. The variable names and the corresponding filenames in the data products are listed in Table A3.





**Table A1.** Sentinel-3 SL_1_RBT files and variables used. Here [X] denotes the SLSTR band number $1 - 6$.

| Variable name | Variable |
|---|---|
| File: geodetic_an.nc | |
| latitude_an | Latitude of detector FOV centre on the Earth's surface, nadir view |
| longitude_an | Longitude of detector FOV centre on the Earth's surface, nadir view |
| File: geodetic_ao.nc | |
| latitude_ao | Latitude of detector FOV centre on the Earth's surface, oblique view |
| longitude_ao | Longitude of detector FOV centre on the Earth's surface, oblique view |
| File: geodetic_tx.nc | |
| latitude_tx | Latitude of detector FOV centre on the Earth's surface |
| longitude_tx | Longitude of detector FOV centre on the Earth's surface |
| File: geometry_tn.nc | |
| solar_zenith_tn | Solar zenith angle, nadir view |
| File: geometry_to.nc | |
| solar_zenith_to | Solar zenith angle, oblique view |
| File: SXX_radiance_an.nc | |
| S[X]_radiance_an | TOA radiance for channel S[X] (A stripe grid, nadir view) |
| File: S[X]_quality_an.nc | |
| S[X]_solar_irradiance_an | Solar irradiance at top of atmosphere, channel S[X], nadir view |
| File: S[X]_radiance_ao.nc | |
| S[X]_radiance_ao | TOA radiance for channel S[X] (A stripe grid, oblique view) |
| File: S[X]_quality_ao.nc | |
| S[X]_solar_irradiance_ao | Solar irradiance at top of atmosphere, channel S[X], oblique view |

**Appendix B: Input and output variables of the models**

We divide the input and output variables into following five groups.

**Geometry variables**

– SYN_altitude

– SYN_O_VAA





**Table A2.** Sentinel-3 OL_1_ERR files and variables used. Here [YY] denotes the OLCI band number $1 - 21$.

| Variable name | Variable |
|---|---|
| File: geo_coordinates.nc | |
| latitude | DEM corrected latitude |
| longitude | DEM corrected longitude |
| File: qualityFlags.nc | |
| quality_flags | Classification and quality flags |
| File: instrument_data.nc | |
| detector_index | Detector index |
| solar_flux | In-band solar irradiance, seasonally corrected |
| File: tie_geometries.nc | |
| SZA | Solar zenith angle |
| File: Oa[YY]_radiance.nc | |
| Oa[YY]_radiance | TOA radiance for OLCI acquisition band Oa[YY] |

– SYN_O_VZA

– SYN_O_SAA

– SYN_O_SZA

– SYN_SN_VAA

– SYN_SN_VZA

– SYN_SO_VAA

– SYN_SO_VZA

– SYN_O_scattering_angle

– SYN_SO_scattering_angle

– SYN_SN_scattering_angle

Here all variables are based on the Sentinel-3 Synergy data product. SYN_O, SYN_SN and SYN_SO correspond
to OLCI, SLSTR nadir view and SLSTR oblique view, respectively.

**Satellite observation variables**

– SL1_S1_reflectance_nadir



**Table A3.** Sentinel-3 SY_2_SYN files and variables used.

| Variable name | Variable |
|---|---|
| File: time.nc | |
| start_time | Time of start measurement |
| stop_time | Time of stop measurement |
| File: geolocation.nc | |
| altitude | DEM corrected altitude |
| lat | DEM corrected latitude |
| lon | DEM corrected longitude |
| File: Syn_AMIN.nc | |
| AMIN | Aerosol Model Index Number |
| File: Syn_Angstrom_exp550.nc | |
| A550 | Aerosol Angstrom exponent at 550 nm |
| File: Syn_AOT550.nc | |
| T550 | Aerosol optical thickness |
| T550_err | Aerosol optical thickness standard error |
| File: flags.nc | |
| SYN_flags | Synergy classification and aerosol retrieval flags |
| CLOUD_flags | Synergy cloud flags |
| OLC_flags | Selected quality and classification flags for OLCI SYN channels |
| SLN_flags | Exception summary and confidence flags for SLSTR nadir-view SYN channels |
| SLO_flags | Exception summary and confidence flags for SLSTR oblique-view SYN channels |
| File: Syn_Oa[XX]_reflectance.nc | |
| SDR_Oa[YY] | Surface directional reflectance associated with OLCI channel [XX] |
| SDR_Oa[YY]_ERR | Surface directional reflectance error estimate associated with OLCI channel [XX] |
| File: Syn_S[YY]N_reflectance.nc | |
| SDR_S[YY]N | Surface directional reflectance associated with SLSTR channel [YY] acquired in nadir view |
| SDR_S[YY]N_ERR | Surface directional reflectance error estimate associated with SLSTR channel [YY] acquired in nadir view |
| File: Syn_S[YY]O_reflectance.nc | |
| SDR_S[YY]O | Surface directional reflectance associated with SLSTR channel [YY] acquired in oblique view |
| SDR_S[YY]O_ERR | Surface directional reflectance error estimate associated with SLSTR channel [YY] acquired in oblique view |
| File: tiepoints_olci.nc | |
| OLC_TP_lat | Latitude (WGS-84) |
| OLC_TP_lon | Longitude (WGS-84) |
| OLC_VAA | OLCI view azimuth angle |
| OLC_VZA | OLCI view zenith angle |
| SAA | Sun Azimuth Angle |
| SZA | Sun Zenith Angle |
| File: tiepoints_slstr_n.nc | |
| SLN_TP_lat | Latitude (WGS-84) |
| SLN_TP_lon | Longitude (WGS-84) |
| SLN_VAA | SLSTR nadir view azimuth angle |
| SLN_VZA | SLSTR nadir view zenith angle |
| File: tiepoints_slstr_o.nc | |
| SLO_TP_lat | Latitude (WGS-84) |
| SLO_TP_lon | Longitude (WGS-84) |
| SLO_VAA | SLSTR oblique view zenith angle |
| SLO_VZA | SLSTR oblique view zenith angle |
| File: tiepoints_meteo.nc | |
| air_pressure | Mean air pressure at sea-level |
| ozone | Total columnar ozone |
| water_vapour | Total column water vapour |





- – SL1_S1_reflectance_oblique

- – SL1_S2_reflectance_nadir

– SL1_S2_reflectance_oblique

- – SL1_S3_reflectance_nadir

- – SL1_S3_reflectance_oblique

- – SL1_S4_reflectance_nadir

- – SL1_S4_reflectance_oblique

– SL1_S5_reflectance_nadir

- – SL1_S5_reflectance_oblique

- – SL1_S6_reflectance_nadir

- – SL1_S6_reflectance_oblique

- – OL1_Oa01_reflectance

– OL1_Oa02_reflectance

- – OL1_Oa03_reflectance

- – OL1_Oa04_reflectance

- – OL1_Oa05_reflectance

- – OL1_Oa06_reflectance

– OL1_Oa07_reflectance

- – OL1_Oa08_reflectance

- – OL1_Oa09_reflectance

- – OL1_Oa10_reflectance

- – OL1_Oa11_reflectance

– OL1_Oa12_reflectance

- – OL1_Oa13_reflectance





- OL1_Oa14_reflectance

- OL1_Oa15_reflectance

- OL1_Oa16_reflectance

– OL1_Oa17_reflectance

- OL1_Oa18_reflectance

- OL1_Oa19_reflectance

- OL1_Oa20_reflectance

- OL1_Oa21_reflectance

**SYN L2 variables**

- SYN_AOD550

- SYN_AOD550err

- SYN_AE550

- SYN_AMIN

– SYN_SYN_no_slo

- SYN_SYN_no_sln

- SYN_SYN_no_olc

- SYN_SDR_Oa01

- SYN_SDR_Oa02

– SYN_SDR_Oa03

- SYN_SDR_Oa04

- SYN_SDR_Oa05

- SYN_SDR_Oa06

- SYN_SDR_Oa07

– SYN_SDR_Oa08





- SYN_SDR_Oa09

- SYN_SDR_Oa10

- SYN_SDR_Oa11

- SYN_SDR_Oa12

– SYN_SDR_Oa16

- SYN_SDR_Oa17

- SYN_SDR_Oa18

- SYN_SDR_Oa21

- SYN_SDR_S1N

– SYN_SDR_S1O

- SYN_SDR_S2N

- SYN_SDR_S2O

- SYN_SDR_S3N

- SYN_SDR_S3O

– SYN_SDR_S5N

- SYN_SDR_S5O

- SYN_SDR_S6N

- SYN_SDR_S6O

**Regression output variables**

– AERONET_AOD_550nm_mean

- AERONET_AOD_440nm_mean

- AERONET_AOD_500nm_mean

- AERONET_AOD_675nm_mean

- AERONET_AOD_870nm_mean



**Correction output variables**

    – AOD550_approximationerror

    – AOD440_approximationerror

    – AOD500_approximationerror

    – AOD675_approximationerror

    – AOD870_approximationerror

Approximation error variables ($\epsilon$) are computed using the Equation 3.

**Inputs and outputs**

As the inputs for the regression model we use the variables from the following variable sets:

    – Geometry variables

    – Satellite observation variables

As the outputs for the regression model we use the variables from the following variable sets:

    – Regression output variables

As the inputs for the correction model we use the variables from the following variable sets:

    – Geometry variables

    – Satellite observation variables

    – SYN L2 variables

As the outputs for the correction model we use the variables from the following variable sets:

    – Correction output variables

**Appendix C: AERONET data used**

The following variables of the AERONET data were used

    – AOD_440nm

    – AOD_500nm



    – AOD_675nm

    – AOD_870nm

465    – 440-870_Angstrom_Exponent

*Author contributions.* AL, JR, AV, HT, TL, and VK developed the deep learning methodology presented. AL collected and processed the data. All authors participated in the data analysis of the results. VK wrote the original manuscript. All authors reviewed and edited the manuscript.

*Competing interests.* The authors declare that they have no conflict of interest.

*Acknowledgements.* This study was funded by the European Space Agency EO science for society programme via POPCORN project. The research was also supported by the Academy of Finland, the Finnish Center of Excellence of Inverse Modeling and Imaging (project 336791) and Academy of Finland (project 321761).



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





**Figure 4.** Estimated AODs at the wavelengths employed in the AERONET. Top to bottom: 440nm, 500nm, 675nm and 870nm.

Left: Sentinel-3 level-2 Synergy AOD product. Middle: Fully learned regressor model. Right: Post-process correction.





**Figure 5.** Rows from top to bottom: AOD (550nm), AE, AI. Left: Sentinel-3 level-2 Synergy product. Middle: Fully learned regressor model. Right: Post-process correction model.





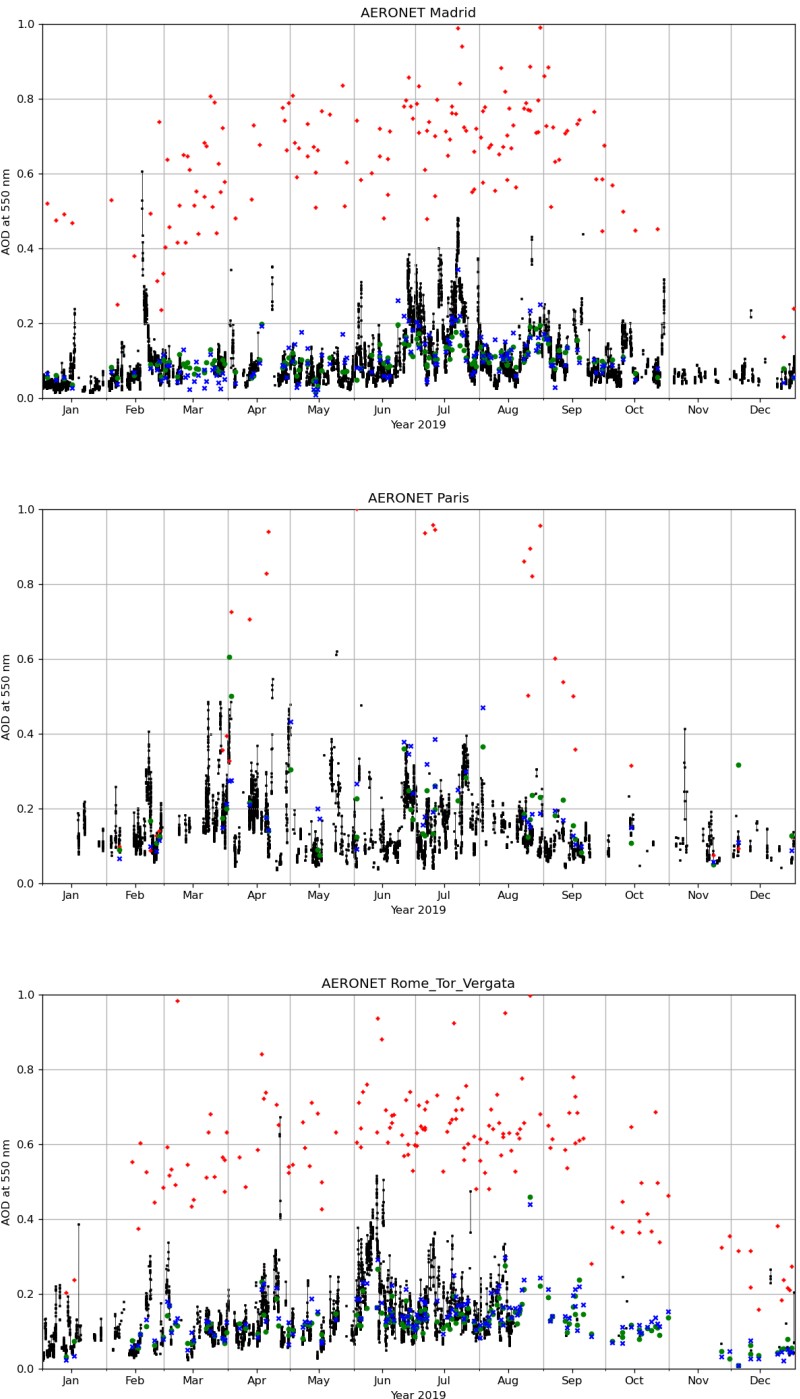

**Figure 6.** AOD at 550 nm time series for three AERONET stations. The black lines and dots indicate AERONET measurements, red diamonds indicate Sentinel-3 level-2 Synergy, green circles regression model, and blue crosses corrected Sentinel-3 Synergy retrievals.





**Figure 7.** Monthly averages of AOD at 550 nm for January (1st row), April (2nd row), July (3rd row), and October (4th row) 2019. Left column: Sentinel-3 level-2 Synergy. Middle column: Regressor model. Right column: Corrected Sentinel-3 Synergy.





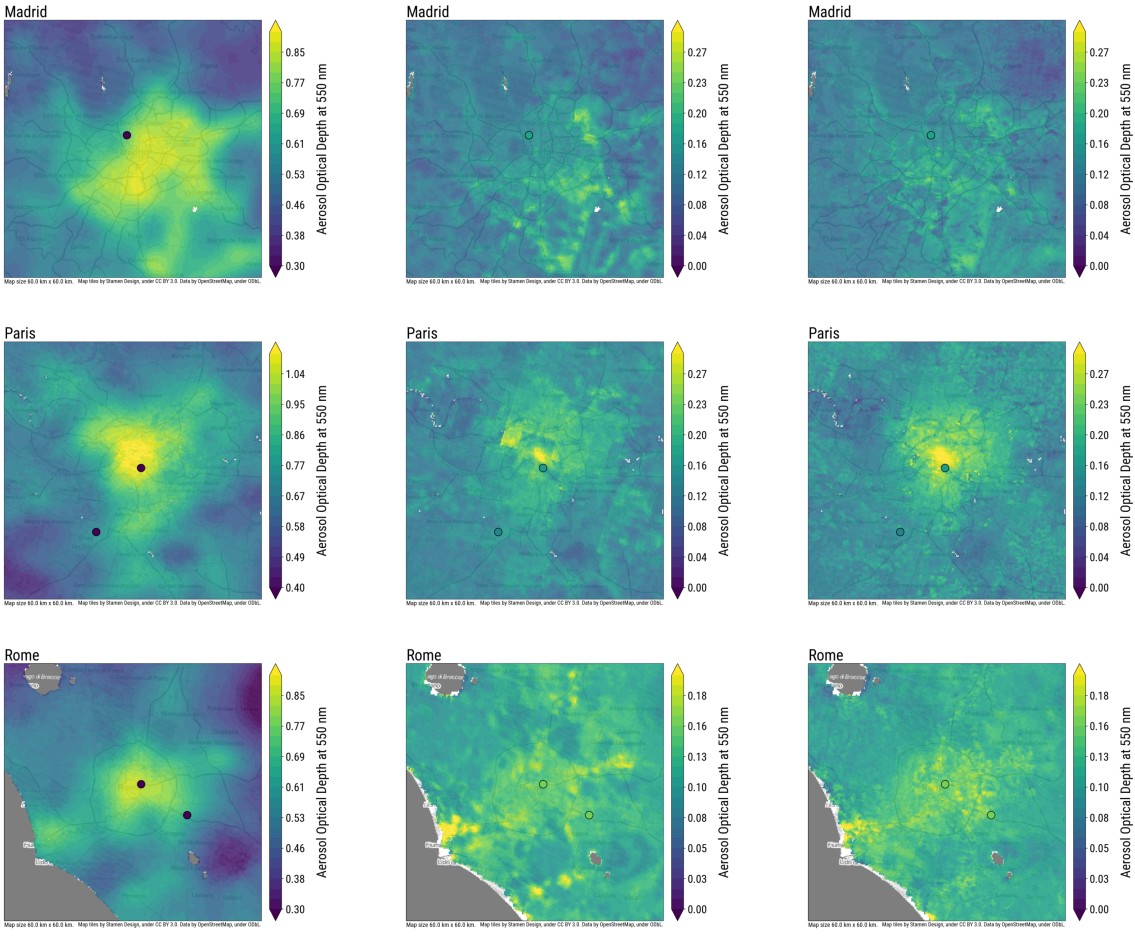

**Figure 8.** July 2019 monthly averages of AOD at 550 nm for Madrid (1st row), Paris (2nd row) and Rome (3rd row). Left column: Sentinel-3 level-2 Synergy. Middle column: Regressor model. Right column: Corrected Sentinel-3 Synergy. Circles represent the monthly averages of AERONET stations.