# Peer review of "Deep Learning Based Post-Process Correction of the Aerosol Parameters in the High-Resolution Sentinel-3 Level-2 Synergy Product"

_Atmospheric Measurement Techniques, 2021_

## Author Comment (AC1)

We would like to thank the reviewer for the comments. Below, we address the comments by the reviewer. The reviewer comments are typed in bold font and our replies to them in regular font. To help the reviewer, we also list some parts of the revised manuscript in our replies and these parts are typed in italic font or with quotation marks for small comments.

In addition to corrections according to referee comments, we have also made some minor (e.g. typos, grammar) related changes to the manuscript.

**Referee 1**

**This paper provides a bias-correction to the Sentinel 3 synergy aerosol data. The method is based on a previously (and recently) published approach by the authors, Lipponen et al (2021), though I feel there is enough difference here to warrant a new publication. The previous application was to MODIS aerosol retrievals; the current work is to a higher resolution product (300 m) for which (I believe) the primary purpose is atmospheric correction for land cover retrievals. Atmospheric correction-based aerosol products are traditionally worse than aerosol-focused aerosol products as in surface-focused cases the atmospheric parameters are often used as an error sink. So doing a bias-correction of them is useful in that it provides a finer resolution aerosol data set than typically available from atmosphere-focused products (which are typically more spatially aggregated in the level 1 to level 2 stage). The bias correction is done using machine learning; a comparison is also made to a fully learned (i.e. level 1 to level 2) machine learning approach. Correcting for the "approximation error" (retrieval error) is expected to be better than a fully-learned approach as the former gains some benefit from retrieval skill (there is less to learn) and that is reasonably borne out by the results presented here. Having a finer spatial resolution is beneficial for eventual air quality applications.**

**The paper is well-written and in scope for the journal. I believe this, combined with Lipponen et al (2021), provide enough evidence that the technique is in principle generalizable. This is important as it implies a fast bias correction could be done for many data sets, which is better for most downstream applications. The authors mention air quality though this is also important for data assimilation which ideally needs unbiased inputs with understood uncertainty characteristics. I don't have any major concerns with what is presented here, and so recommend publication following minor revisions. I would be happy to review the revision if the Editor wishes. I applaud the authors for noting that the code will be available, as this can help speed uptake and transparency is in general a good thing. Hopefully it will be available by the time the final version of this paper is published such that it can be linked directly. I also downloaded the animation linked in the paper and confirm that works and is useful. I will note I am not a machine learning expert so have not commented on the details of that; I recommend at least one reviewer should have machine learning expertise in order to judge that aspect.**

We thank the referee for the encouraging and positive feedback. We have prepared and published in GitHub a code package to post-process correct Sentinel-3 data. The link to this code repository is in the "code and data availability" section of the manuscript. We also added link to the post-process corrected Sentinel-3 aerosol data for year 2019 and all regions of interest.

**My specific comments are as follows:**

1. **Throughout, the authors cite Lipponen et al (2020) for their prior work; this appears to be the preprint of the final Lipponen et al (2021) paper describing this technique applied to MODIS. I assume this is an oversight but it should be corrected.**

Thank you for noting. Reference to Lipponen et al (2021) was updated.

2. **Section 3.1: The authors link to the ESA website to describe the Sentinel 3 source retrievals. Are there no publications or tech documents that can be cited here? The linked page is not informative (it's basically to a catalogue of products, no ATBD or validation report etc). I would like to know a bit about the general Sentinel 3 synergy algorithm approach, e.g. how the SLSTR and OLCI measurements (with different pixel sizes) are used and combined, what the main assumptions are (it looks from Figure 5 like a fixed value of AE is used, for example). The Conclusions notes that the standard synergy aerosol product is at 4.5 km but it is not clear to the reader why, especially if this is primarily an atmospheric correction algorithm which is normally done at fine resolution, and the SLSTR data are 1 km or finer – do they do the atmospheric correction at coarser scale than the surface retrieval? Or are the "land" and "aerosol" synergy products entirely separate? I know this is not the authors' algorithm but presumably the synergy product is not a mystery black box (someone somewhere knows what the algorithm is) and as an ESA product this information should be available to the public somewhere such that a summary can be given here. If not then please point this comment to the responsible ESA official because there really needs to be some documentation for a data product if it is put out to the public. It is all frustratingly opaque and, after clicking around the ESA site for some time, I was unable to satisfy my curiosity.**

According to referee's suggestion the Synergy algorithm ATBD was cited. The publicly available ATBD, however, is quite old and corresponds to the older version of the algorithm than the operational algorithm. New version of the ATBD does not seem to be publicly available. We have given feedback to ESA for easier access of the information about the Synergy algorithm technical details.

The spatial resolution difference in land and aerosol AOD is due to entirely separate land and aerosol Synergy data products. In our manuscript, we only consider the higher resolution "land" AOD.

3. **Section 3.2: I believe the preferred citation for AERONET version 3 direct Sun is Giles et al (2019): https://amt.copernicus.org/articles/12/169/2019/ This should be given in addition to or instead of Holben et al (1998).**

Yes, we have now also cited the Giles et al. (2019) for a reference to the latest AERONET version 3 Direct Sun.

4. **Section 3.3: elsewhere in the paper the authors (rightly) note that some previous machine learning studies give an artificially high impression of performance by not having independent training and validation data sets. In this section the authors note that they split training/validation data by station, which is better than splitting individual observations within individual stations. I agree with this. However, it does seem a bit of a missed opportunity not to test the approach on something fully outside of the selected regions of interest, and more fully independent from the training set. Figure 2, for example, reveals many sites (individual or clustered) outside of these regions. I suggest the authors extend their validation to a few of these "untrained" sites or regions to see what the benefit of the networks is there – this will provide more evidence for how applicable the model is on a global scale with limited regional training. I know data volume is limited considering only 1 year of data but hopefully we can say something at least. I would suggest looking at sites in Amazonia (contrast between clear seasons and heavy biomass burning, in a somewhat cloudy environment), Korea/Japan (mixed aerosol types, good AERONET site density), and/or Australia (traditionally a difficult area for aerosol retrievals). The paper is not too**

**long and I think adding this would add substantial further interest to the reader without making the length excessive. The Korea example dovetails well with my final point below.**

Thank you for the very good comment regarding the results completely outside the training data set. We agree that ideally, we could consider looking at the regions listed by the referee. However, we do not have the data available and the official Copernicus Open Access hub sharing Sentinel-3 data has already archived the data for year 2019 so it would be very tedious to get the analysis done. Instead, we have carried out an additional analysis in which we evaluate the generalization capabilities of our approaches in Central Europe. In this additional analysis, we take the training data from all other regions of interest (Eastern USA, Western USA, Southern Africa, India), train the fully learned and post-process correction models and apply these models to Central Europe data. Please find the results in the figure below.

[Figure]

Figure 1. AOD (550 nm) for Central Europe and year 2019. Machine learning models are trained using data outside Central Europe region. Left: Sentinel-3 level-2 Synergy product. Middle: Fully learned regressor model. Right: post-process correction.

We have added this figure to the manuscript and added the following paragraph to the Results section of the manuscript:

> *To study the generalization capabilities of the models, we carried out a test in which we evaluated the fully learned and post-process correction models' accuracy in the Central Europe region. The machine learning models were trained using data from regions of interest outside Central Europe (Eastern USA, Western USA, Southern Africa, India). The test aimed to evaluate how the models generalize to data far from the training data regions, possibly with different dominant aerosol types and surface reflectances. Figure 1 [Fig. 1 in this reply to referees, Fig. 9 in the manuscript] shows the results for this test for the AOD at 550 nm in the Central Europe region. The post-process correction results in clearly more accurate AOD estimates than the fully learned model. The result indicates that using the training data from nearby regions improves the model performance, and the post-process correction model performs better than the fully learned model also in regions far from the training data regions.*

We also added the following paragraph in the conclusions:

> *We also studied the generalization capabilities of the machine learning models. The results show that the post-process correction model performs better than the fully learned model also when trained using data from distant regions. Ideally, in an operational setting, the machine learning models would be trained using global data, but, for example, in AOD retrievals, regardless of the*

*high number of AERONET stations, there are always some regions with a relatively poor AERONET coverage. Therefore, based on our results, we expect the post-process correction method to perform better than the fully learned models in these regions.*

5. **Figure 8: It is ok to have the scale different for each row because each region is quite different. But I think the scale for each panel in a given row should be made the same, for more direct comparability. I acknowledge that the scale is quite different because the uncorrected synergy product is a lot higher than the others, but if a logarithmic scale were used (as in figure 7) I feel the plots would be better without loss of contrast within and between them.**

Figure 8 was improved according to the referee's suggestion – now each row (city) has the same logarithmic color scale. Also, the colormap was changed to the same as used in Figure 7.

6. **Conclusions: the authors' fully-learned and bias-corrected approaches clearly work better than the standard synergy aerosol data at the AERONET sites. The regional maps also look more reasonable. But there is an uncertain middle ground on the scales of a few to tens of km. It's hard to know whether some of the fine structure in these maps is real variation, statistical noise, or surface-related artefacts. For example, returning again to Figure 8, there are AOD hotspots corresponding to the built-up locations. With only 1 or 2 AERONET sites in each area, how are we to know if this spatial structure is real? This is not a problem the authors can fully solve but it is something that should be acknowledged. I know there have been some regionally-dense AERONET deployments (dozens of sites in a comparable region); Korus-AQ (summer 2016) was early in the Sentinel 3 era with 20 AERONET sites (Choi et al 2021: https://doi.org/10.1016/j.atmosenv.2021.118301 ), maybe that could be looked at (here or elsewhere). There is also a network of shadowband radiometers providing aerosol properties distributed around the Southern Great Plains ARM site region in the USA which I believe were operational during 2019 (https://www.arm.gov/capabilities/instruments/mfrsr ). For me this "variability on tens of km" scales is the key next step we need to solve as we move toward better fine-scale aerosol retrievals. In addition to expanding the text to draw more attention to this issue (which may attract further studies/funding on the problem) I encourage the authors to expand the paper by looking at one or both of these areas, if data are available, to take a first step.**

We agree with the referee that it is very difficult to evaluate the accuracy of the high-resolution features of AOD in the presence of only sparse AERONET measurements. We have acknowledged this in the conclusions by adding the following paragraph:

*We acknowledge the difficulty in validating the high spatial resolution satellite aerosol data products as accurate high-resolution spatial coverage aerosol validation data does not exist. There are, however, some ground-based and aircraft measurement campaigns such as Distributed Regional Aerosol Gridded Observations Network (DRAGON) (Garay et al., 2017; Virtanen et al., 2018, e.g.), KORea–United States Air Quality (KORUS-AQ) (Choi et al., 2021, e.g.), and the Atmospheric Radiation Measurement (ARM) program (Javadnia et al., 2017, e.g.) that could provide helpful insight on high-resolution aerosol features. Using the campaign data from these campaigns to validate the high-resolution satellite aerosol retrievals is a potential topic for future studies.*

---

## Author Comment (AC2)

We would like to thank the reviewer for the comments. Below, we address the comments by the reviewer. The reviewer comments are typed in bold font and our replies to them in regular font. To help the reviewer, we also list some parts of the revised manuscript in our replies and these parts are typed in italic font or with quotation marks for small comments.

In addition to corrections according to referee comments, we have also made some minor (e.g. typos, grammar) related changes to the manuscript.

**Referee 2**

This article applied a previous developed concept of using machine learning (ML) to biascorrect aerosol optical depth (AOD) and other aerosol data from conventional aerosol product. Original concept of ML post-processing of satellite data against ground truth is introduced in author's previous journal articles. This time a feed forward neural network is used on Sentinel-3 data to produce two aerosol products: machine learning generated aerosol data and bias-corrected Level-2 Synergy Product. The article claims that the postprocess corrected the Sentinel-3 synergy product is a high resolution, better accuracy data products than the original aerosol product and the aerosol product generated from pure FFNN model. Within resent decade, machine learning has been rapidly applied to Earth Science field. One of doubtfulness of relying on ML is that the approach is not based on physics. The idea of machine learning post-process include both the state of art machine learning technique and traditional algorithm-based approach, which maintain the physics within the retrieval process. It is a conservative way of using ML and if successful, can be applied to many fields. However, the statement of the post-process corrected aerosol data has higher accuracy than full ML predicted aerosol data is not convincing, especially in terms of AOD. Figure 4, 5, and 6 all show comparisons between these two products. There is no significant improvement from post-process corrected product to full machine learning output. Although the error statistics against AERONET are slightly better in post-process corrected data, when investigate details in Figure 4 we can see that the overestimation of AOD especially at AOD < 0.2, is amplified in post-process corrected data than fully learned regressor model output. The smaller bias statistics in post-processed product is balanced by the overestimation in low AOD regime (AOD < 0.2) and underestimation in high AOD regime (AOD > 0.5). If we look at other evaluation plots, such as error histogram or error diagnostic plot. We may have much better look at the error distribution of two data sets. Similarly for AE comparisons, it is hard to say that the accuracy of AE prediction is improved between the two ML-involved products.

We thank the referee for the careful evaluation of our manuscript and the comments.

We kindly disagree with the referee's statement "There is no significant improvement from postprocess corrected product to full machine learning output." At first, the absolute improvements may not seem significant. However, the relative improvement, for example, in AOD at 550 nm is significant (R2 improves by about 9%, RMSE is about 8% smaller, and BIAS decreases by 20% in post-process corrected model when compared to fully learned model). In some applications, such as data assimilation, these relative improvements may be significant for the accuracy of the data assimilation model. The referee also claims that "...when investigate details in Figure 4 we can see that the overestimation of AOD especially at AOD < 0.2, is amplified in post-process corrected data than fully learned regressor model output.". This claim is not true. The biases for AERONET AOD smaller than 0.2 and larger than 0.5 are shown in the tables 1 and 2 below. The post-process corrected AOD has the best bias metric for all wavelengths (best model shared with the fully learned model in 3 cases) and thus the data does not support the referee's claim.

| Wavelength | Synergy AOD bias | Fully learned AOD | Post-process       |
|------------|------------------|-------------------|--------------------|
|            |                  | bias              | corrected AOD bias |
| 440 nm     | 0.380            | 0.011             | 0.011              |
| 500 nm     | 0.333            | 0.010             | 0.010              |
| 550 nm     | 0.303            | 0.010             | 0.009              |
| 675 nm     | 0.249            | 0.008             | 0.008              |
| 870 nm     | 0.188            | 0.007             | 0.006              |

Table 1. AOD biases corresponding to data points with AERONET AOD smaller than 0.2. The graybackground indicates the best-performing model.

| Table 2. AOD biases corresponding to data points with AERONET AOD larger than 0.5. The | e gray |
|----------------------------------------------------------------------------------------|--------|
| background indicates the best-performing model.                                        |        |

| Wavelength          | Synergy AOD bias    | Fully learned AOD | Post-process       |
|---------------------|---------------------|-------------------|--------------------|
|                     |                     | bias              | corrected AOD bias |
| 440 nm              | 0.484               | -0.294            | -0.271             |
| 500 nm              | 0.417               | -0.267            | -0.245             |
| 550 nm              | 0.379               | -0.243            | -0.222             |
| 675 nm              | 675 nm 0.299 |                   | -0.175             |
| 870 nm 0.247 |                     | -0.137            | -0.122             |

We added the following paragraph to the results section:

To evaluate the models' performance in low and high AOD conditions, we evaluated the results corresponding to AERONET AOD at 550 nm smaller than 0.2 and larger than 0.5. The results are shown in Table 1 [of the manuscript]. The post-process corrected model results in the best bias metric in both low and high AOD conditions. In addition, the post-process corrected model results in the best R2 in low AOD and the best RMSE in high AOD conditions. The fully learned model results in about 4 % lower RMSE than the post-process corrected model in small AOD. The Synergy R2 is the best for the high AOD cases but there are only 163 samples in the high AOD cases so more data would be needed for more reliable evaluation of the models in high AOD conditions.

We also added the following table of the results for low and high AOD in the manuscript:

| AOD 550 nm $< 0.2$ (N=4708) |         |               |                        |  |  |
|-----------------------------|---------|---------------|------------------------|--|--|
| Metric                      | Synergy | Fully learned | Post-process corrected |  |  |
| $R^2$                       | 0.113   | 0.270         | 0.310                  |  |  |
| RMSE                        | 0.412   | 0.050         | 0.052                  |  |  |
| Bias                        | 0.303   | 0.010         | 0.009                  |  |  |
| AOD 550 nm > 0.5 (N=163)    |         |               |                        |  |  |
| Metric                      | Synergy | Fully learned | Post-process corrected |  |  |
| $R^2$                       | 0.497   | 0.273         | 0.377                  |  |  |
| RMSE                        | 0.433   | 0.313         | 0.279                  |  |  |
| Bias                        | 0.379   | -0.243        | -0.222                 |  |  |

Table 1. Error metrics for the satellite data product AOD at 550 nm corresponding to small (<0.2) and large (>0.5) AERONET AOD. The bold font indicates the best performing model.

**Other specific comments are:**

**Line 27, atmospheric spelled wrong.**

Corrected.

**Line 67 remove "accurate"**

Removed.

**Line 107 In section ? missing a number.**

Corrected. "In section 2,..."

**Line 190 please specific list the time/spatial criteria for collocation.**

The temporal and spatial collocation is now better described. The sentence citing Petrenko et al. (2012) was revised to: "We use the same  $\pm 30$  minutes temporal thresholds for the collocation procedure as in Petrenko et al. (2012) and spatial collocation radius of 5 km."

**Line 197-198 Can random split for each region result in data from a few sites dominate the results for one region?**

We have tested how the random split affects the results by running the analyses with multiple different random splits. As there are quite many stations in each region of interest there are no single station that would dominate the results and therefore different random splits do not significantly change the results. To show this result to the readers we have added the following sentences to the manuscript: "To study the effect of randomness on the splits of AERONET stations, we tested our approach with multiple random splits. We did not observe significant differences in the results between different random splits of the AERONET stations."

**Line 211-212 Regarding normalization method. If we use all data mean/std to do the z-score standardization, all the data is converted equally still within the same scale as they are originally. What is the point of normalization? For fill data, what average is used? and how much missing data is there?**

The normalization is often used in machine learning to ensure we do not run into numerical problems due to input values of different orders of magnitudes. Large differences in the values of the data may cause numerical problems in the training or evaluation of the neural network. This is the reason we carried out the normalization.

The missing values were filled with the average value of the corresponding variable in the training data set (in the manuscript: "In case some of the inputs contains a missing value, it is filled with the average value of the training dataset.")

Most of the missing values were due to different swath widths of OLCI, SLSTR nadir and oblique views. On average, there were about 8 % and 6 % missing values in the fully learned model and post-process correction model datasets, respectively. We added the following sentence to the manuscript: "On average the input data of the fully learned and post-process correction models contained about 8 % and 6 % of missing values, respectively."

**Section 3.5 What is the accuracy for the two-folds testing results for training/testing/validation datasets?**

As mentioned in the manuscript we have split the data into two sets by random selection of AERONET stations. In the evaluation, the models are always trained using the other set and evaluated using the other. In the training of the neural network models, we use, according to proper machine learning practices, early truncation based on monitoring of the validation loss to avoid overfitting. The accuracy metrics for the data computed using the models trained on the same data are significantly better than the ones computed for the stations not included in the training data. This is expected and well-known behavior in machine learning and should be avoided. We think it is not informative to present the overoptimistic results that contain evaluation data corresponding to models trained with same data. To get an idea of this type of evaluation results for AOD at 550 nm obtained with models trained on the same data see the figure below.

Figure 2. AOD (550nm). Left: Sentinel-3 level-2 Synergy product. Middle: Fully learned regressor model trained. Right: Post-process correction model. Please note the models have been evaluated using the training datasets and thus do not represent the true error metric values.

---

## Author Response (AR2)

Below, we address the comments by the editor. The editor comments are typed in bold font and our replies to them in regular font.

**Referee # 1:**
**Last comment: Authors may want to add ORCALES (https://espo.nasa.gov/ORACLES/content/ORACLES) in the list of field campaigns. The HSRL-2 and 4STAR made high-res. measurements of AOD over the Southeastern Atlantic Ocean in the Aug-Sep-Oct of 2016, 2017, & 2018. Also, how about comparing the post-process bias corrected Sentinel-3 product against the 1-km AOD retrieval dataset from MAIAC? Although, the exercise won't constitute a validation of either product, but still can be useful to evaluate the relative difference, especially the spatially varying AOD features.**

We have added a mention about the ORACLES campaign in the list of field campaigns. Thank you also for mentioning the HSRL-2 and 4STAR high-res. AOD measurement. As they are over ocean and we have only used data over land we did not add them to this manuscript. We also now mention the Synergy - MAIAC comparison as an interesting and potentially useful comparison to reveal information about spatially varying AOD features. However, such comparison with MAIAC and high-resolution campaign data such as DRAGON is left for future studies.

**Referee # 2:**
**I agree with the anonymous referee # 2 that the post-process corrected product is marginally better (not significant), in terms of comparison statistics (R2, RMSE, Bias), than the full ML output product. However, at the same time, I also agree with your results that relatively (%) bias corrected product provides best statistical comparison. Regarding the results tabulated in Table 1 and Table 2, referee # 2 appears to have made a wrong claim as the bias values are lowest in the post-correction AOD dataset. However, the improvement is marginal over the fully learned AOD product.**

**The revised paper, especially abstract and conclusion, should clearly accept and reflect this finding. The benefit and superiority of the proposed post-correction method is that the full processing of the satellite aerosol products is not required, which is a time-consuming effort often carried out by the algorithm developers/processors, and not by the individual researchers. Since authors have made the post-correction package available for the users free of cost, it will facilitate researchers applying the proposed correction without requesting full processing to the algorithm developers.**

We have revised the abstract and conclusions according to your comments. Please see the difference pdf showing the changes made to abstract, introduction, and conclusions.